

# The role of echinacoside-based cross-linker nanoparticles in the treatment of osteoporosis

Dandan Hu[1,*], Chunan Cheng[2,*], Zhen Bian[1] and Yubo Xu[1]

[1] Department of Stomatology, Shanghai East Hospital, School of Medicine, Tongji University, Shanghai, China
[2] Department of Oral and Maxillofacial Surgery, Stomatological Hospital and Dental School of Tongji University, Shanghai Engineering Research Center of Tooth Restoration and Regeneration, Shanghai, China
[*] These authors contributed equally to this work.

## ABSTRACT

**Background.** Current drugs for treating osteoporosis may lead to toxic side effects. Echinacoside (ECH) is a natural small molecule drug. This study examined and compared the therapeutic effects of cross-linker (CL)-ECH and ECH-free nanoparticles on osteoporosis.

**Methods.** Echinocandin-based CL-ECH nanoparticles were prepared, and the nanoparticle size and drug loading were optimized and characterized by adjusting the ratio. The antioxidant effect of CL-ECH nanoparticles on bone marrow-derived macrophages (BMDMs) was analyzed using flow cytometry, immunofluorescence staining and quantitative real-time polymerase chain reaction (qRT-PCR). Bone marrow stromal cells (BMSCs)-based detection of bone-producing effects was conducted using alkaline phosphatase (ALP), Alizarin Red S (ARS) and qRT-PCR. TRAP, phalloidin staining, and qRT-PCR was performed to detect osteogenesis-inhibiting effect on BMDMs. CL-ECH nanoparticles were applied to treat an ovariectomized (OVX) mouse model at low doses.

**Results.** Compared to ECH, CL-ECH nanoparticles suppressed oxidative stress in BMDMs by promoting NRF-2 nuclear translocation, which inhibited the production of both reactive oxygen species (ROS) and osteoclast production through downregulating NF-$\kappa$B expression, with limited effect on the osteogenesis of BMSCs. *In vivo* studies showed that low-dose CL-ECH nanoparticles markedly improved bone trabecular loss compared to ECH administration in the treatment of osteoporosis.

**Conclusions.** The current discoveries provided a solid theoretical foundation for the development of a new generation of anti-bone resorption drugs and antiosteoporosis drugs.

# INTRODUCTION

Acceleration of global population ageing has increased the burden to prevent and control osteoporosis (*Diab et al., 2019*; *Guo et al., 2023*; *Wang et al., 2023*). The limitations of traditional antiosteoporosis drugs are becoming increasingly apparent. It is predicted that the medical cost of osteoporotic fractures will be as high as USD 163 billion in 2050

Corresponding authors
Zhen Bian, bianzhen.cn@163.com
Yubo Xu,
xuyubo18907087733@163.com

(*Si et al., 2015*). Therefore, a new generation of antiosteoporosis drugs should be developed to improve the current osteoporosis treatment. Research showed that some natural alternatives with fewer adverse side effects could help prevent bone loss and fracture risk related to osteoporosis (*Matzkin et al., 2019*).

Echinacoside (ECH) is a phenylethanol glycoside with antioxidant and anti-inflammatory properties (*Song et al., 2021*). In a rat spinal cervical spondylosis model, it has been found that ECH could reduce the inflammatory response by inhibiting excessive mitochondrial division and lowering reactive oxygen species (ROS) production, which ultimately suppresses the secretion of inflammatory factors such as interleukin (IL)-6, IL-1 and tumor necrosis factor (TNF)-α (*Zhou et al., 2020*). The negative effects of excessive oxidative stress on bone metabolism have been analyzed (*Agidigbi & Kim, 2019*; *D'Ambrosio et al., 2019*). In addition, studies have also elucidated the protective effect of ECH using animal models of ovariectomized (OVX) osteoporosis (*Li et al., 2013*; *Yue et al., 2022*). Normally, ECH are orally administrated, with a treatment period up to 12 weeks. ECH is a water-soluble drug but its bioavailability of oral administration is low and it metabolizes quickly, therefore, to maintain a certain level of drug concentration in the body, ECH needs to be administrated at a high dose (*Liu et al., 2018*; *Tian et al., 2021*; *Wu et al., 2020*). However, long-term dosing of high concentration of the acting drug in the body will result in greater toxic side effects on other tissues and organs (*He et al., 2019*).

The use of cross-linker (CL) pellets has been investigated in a variety of osteogenesis studies (*Chiu et al., 2012*; *Hachinohe et al., 2022*). To reduce possible side effects of ECH, we constructed a slow-releasing system with CL pellets as a simple and effective carrier for small doses of intravenous injection. In addition, we compared the CL-ECH nanoparticles and the use of only ECH to analyze their antioxidant effect on bone marrow-derived macrophages (BMDMs), osteogenic ability in bone marrow stromal cells (BMSCs), and their inhibitory effect on osteoclastogenesis, aiming to offer a theoretical basis for osteoporosis treatment with low-dose nanodrugs.

## METHODS

### Reagents

Dulbecco's Modified Eagle Medium (DMEM) and fetal bovine serum (FBS) were commercially purchased from Gibco Inc. (Themo Fisher Scientific, Waltham, MA, USA). Echinacoside, cross-linker, and esterase were purchased from MedChemExpress (Monmouth Junction, NJ, USA), Source Leaf (China), and Maclean (China), respectively. M-CSF and RANKL were acquired from PeproTech Inc. (Themo Fisher Scientific).

### BMDMs and BMSCs

Following the description in a previous study (*Marim et al., 2010*), BMDMs were extracted from the femurs and tibiae of 6- to 7-week-old C57BL/6 mice purchased from Shanghai Laboratory Animal Center (Shanghai, China). DMEM was used to culture the BMDMs, and 10% heat-inactivated FBS and 1% penicillin-streptomycin were added as supplements.

## CL-ECH nanoparticle preparation and characterization

ECH (2 mg) and CL (cat no. S26218; Source Leaf; 1:4; 1:2; and 1:0.5 molar ratios, respectively) were dissolved in dimethyl sulfoxide (DMSO) solvent (100 µL) overnight. The solution was added dropwise to deionized water (five mL) and stirred for 2 h. The CL-ECH nanoparticles were obtained by ultrafiltration (8,000 rpm, 30 min), resuspended in one mL of deionized water, and filtrated through a filter (0.22 µm). The size of the nanoparticles was measured with dynamic light scattering and the morphology was observed under a transmission electron microscopy (TEM). Nanoparticles were measured under an ultraviolet (UV)-visible spectroscopy (Thermo Fisher Scientific, Waltham, MA, USA), and the content of ECH was calculated using the standard curve of ECH (Fig. S1).

After co-culturing the cells with nanoparticles for 4 h, the uptake of Cy5-loaded CL-ECH nanoparticles in BMDMs and BMSCs was detected using flow cytometry (Beckman Coulter, Danaher Corporation, Brea, CA, USA).

## Intracellular ROS measurement

DCFH-DA (20,70-dichlorodihydrofluorescein-diacetate) stock solution (Calbiochem, Germany) was prepared using DMSO and kept in the dark at 20 °C. On culture dishes (60-mm), BMDMs were primed with 50 µM of ECH or CL-ECH (5, 10, 25, 50 µM) for 6 h and then added with 100 ng/mL of LPS (lipopolysaccharides from Salmonella enterica serotype Minnesota; Sigma-Aldrich, St. Louis, MO, USA) when the confluence reached 90%. After 24-hour LPS stimulation, the cells were treated with 10 M of DCFH-DA for 30 min. The green fluorescence of DCFH-DA was recorded at 515 nm with flow cytometry (Beckman Coulter).

## Visualization of NRF2 nuclear translocation with confocal microscopy

BMDMs ($0.1 \times 10^6$) were plated on a 24-well chamber slide and treated by ECH (50 µM) or CL-ECH (10 µM) for 6 h at 37 °C and then by LPS stimulation for 18 h. Next, the cells were fixed with 4% paraformaldehyde and permanently fixed with 0.3% Triton X-100 in phosphate-buffered saline (PBS). The cells were first probed by Anti-pNRF2 primary antibody (DF7519; Affinity) and then by anti-rabbit Alexa Fluor 488 secondary antibody (Life Technologies, Thermo Fisher Scientific). DAPI dye was used to mount the cells, and the cells were photographed using a confocal microscope (Leica, Wetlzar, Germany).

## Real-time polymerase chain reaction (qRT-PCR)

The total RNA was extracted using TRIzol solution (Invitrogen). A Primer Script RT Reagent Kit (Takara Bio, Tokyo, Japan) was used to perform reverse transcription of the RNA samples into complement DNA (cDNA). Three independent PCR amplifications were carried out using RT-PCR system (Roche Diagnostics, Basel, Switzerland) with specific primers and iQ SYBR Green Supermix (Bio-Rad Laboratories, Hercules, CA, USA). The PCR cycling was set at 95 °C for 3 min, followed by 39 cycles at 95 °C for 10 s, at 57 °C for 10 s, and at 72 °C for 30 s.

## ALP staining

After osteogenic differentiation for 7 days, the cells were fixed by 4% polyoxymethylene for 30 min and then washed by PBS 3 times. ALP staining solution (BCIP/NBT alkaline

phosphatase kit; Beyotime, Shanghai, China) was used to incubate with the cells together at 37 °C for 10 min and then the cells were rinsed in distilled water 3 times.

## Alizarin red S staining

After osteogenic differentiation induction, the cells were fixed with 4% polyoxymethylene for 30 min and washed with PBS 3 times. Next, the cells were stained with 1% Alizarin red S (pH 4.2; Sigma-Aldrich, St. Louis, MO, USA) for 15 min. Histomorphometry was observed and photographed using Nikon Eclipse optical microscope (Nikon, Tokyo, Japan) and Nikon D7000 camera (Nikon).

## TRAP and phalloidin staining

BMDMs were plated onto 24-well plates ($5 \times 10^3$ cells/well) in the presence or absence of recombinant RANKL (100 ng/mL) with or without 50 μM ECH or 10 μM of CL-ECH. During 6 days of culture, all media were refreshed every 3 days. The cells were stained for TRAP using an acid phosphatase kit (Sigma-Aldrich) and fluorescein isothiocyanate (FITC) phalloidin (#p5282; Sigma-Aldrich). TRAP-positive, multi-nucleated cells (over 3 nuclei) were defined as dark-red multinucleated cells.

## Hematoxylin and eosin (HE) staining and Masson tissue staining

All the animal experiments were officially approved by the Animal Ethics Committee of Tongji University (Approval number: (2022)- DW- 02) and conducted following the National Institutes of Health guide for the Care and Use of Laboratory Animals. An osteoporotic mouse model was established using 20 eight-week-old female C57BL/6 mice (approximately 25 g, Shanghai Laboratory Animal Center). The mice were randomly divided into 4 groups ($n = 5$) as follows: (I) control group with sham surgery, (II) OVX group, (III) ECH group with free ECH, and (IV) CL-ECH group with CL-ECH nanoparticles. ECH and CL-ECH nanoparticles (400 μg ECH/mL in 0.9% NaCl solution) at a dose of 1 mg/kg were injected into the mice *via* tail vein every 3 days for 8 weeks. Five animals were kept in a cage, fed with standard laboratory food, and provided with access to tap water in a climate-controlled environment (55% humidity, 12 h of light and 12 h of darkness at 25 °C).

The mice were euthanized by cervical dislocation. The femurs were removed bilaterally, and the soft tissues on the surface were collected and then fixed in 4% polyethylene. HE and Masson staining was performed on the tissue sections after decalcification, paraffin embedding, sectioning, and dewaxing.

## Statistical analysis

SPSS 20.0 software (IBM Corp., Armonk, NY, USA) was used for statistical analysis. Data were shown as the mean ± standard deviation. When two independent groups were compared, Student's $t$-test was used. $P < 0.05$ was defined as statistically significant difference.

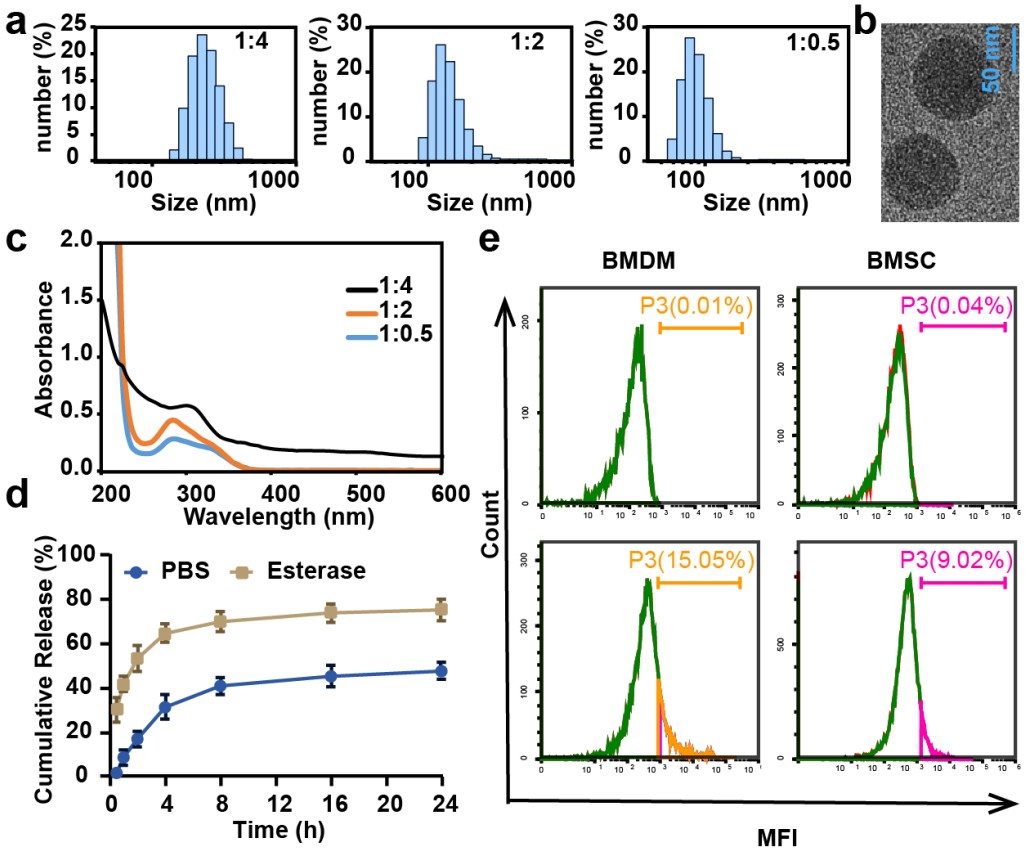

**Figure 1 The characteristics of CL-ECH nanoparticles.** (A) Size distribution of the CL-ECH nanoparticles. (B) TEM image of the nanoparticles. (C) UV-vis spectrum of different ratios of CL-ECH nanoparticles. (D) *In vitro* release profile of the CL-ECH nanoparticles in PBS buffer with or without the esterase. (E) Flow cytometry analyses showing the *in vitro* uptake of Cy5-labeled CL-ECH nanoparticles into BMDMs and BMSCs at 4 h. CL-ECH, cross-linker-echinacoside; TEM, transmission electron microscopy; UV, ultraviolet; PBS, phosphate buffer saline; BMDMs, bone marrow-derived macrophages.

# RESULTS

## Characterization and cellular uptake of CL-ECH

CL-ECH was prepared by mixing cross-linker and ECH at different ratios, and the particle size and UV level were measured. The results showed that the particle size was larger than 100 nm when the ratio was 1:4 and 1:2 and the size became smaller than 100 nm when the ratio was 1:0.5 (Fig. 1A). The TEM results showed a spherical morphology (Fig. 1B). The drug loading rate was measured to be 8.4 weight (wt) % according to the UV results (Fig. 1C) and the standard curve of ECH (Fig. S1). In the presence of esterase, CL-ECH was rapidly released (Fig. 1D).

After 4 h of co-incubation with the cells, flow cytometry analysis showed that Cy5-loaded CL-ECH nanoparticles were endocytosed into BMDMs, with around 15.05% of cells (and 9.02% of BMSC) showing Cy5-positive (Fig. 1E).
## CL-ECH pretreatment promoted the antioxidant effect of BMDMs

### Identification of BMDMs

After induction of mouse monocytes, 97.9% of the cells differentiated into mature macrophages expressing F4/80 marker (Fig. 2A), indicating that the mouse-derived macrophages were successfully cultured and could be used for subsequent experiments.

### ECH or CL-ECH inhibited ROS production in BMDMs

After LPS stimulation, enhanced MFI (*i.e.,* increased ROS production) in BMDMs (Fig. 2A) was detected by flow cytometry. 50-µM ECH significantly reduced ROS levels, and CL-ECH decreased ROS production in a dose-dependent manner. When CL-ECH was 5 µM, ROS was reduced slightly but the difference was not statistically significant in comparison to the LPS group. Similarly, 10-µM CL-ECH also achieved the same effect as 50-µM ECH on reducing ROS. Therefore, 10 µM of CL-ECH was chosen to conduct the subsequent experiments.

### ECH or CL-ECH enhanced the expression of the NRF-2-HO-1 axis

The pNRF-2 is an important transcription factor in the antioxidant stress network. The expression of pNRF-2 was slightly decreased in the LPS group (Fig. 2B) but increased significantly in the 50 µM ECH group and the 10 µM CL-ECH group, with the 10 µM CL-ECH group showing higher increase of pNRF-2 expression.

Compared with the 50 µM ECH group, the expression levels of NRF-2 and HO-1 were significantly higher in the 10 µM CL-ECH group (Fig. 2C).

## CL-ECH and ECH had similar osteogenic effects on BMSC cells

ALP staining and ARS staining in the CL-ECH group showed similar results when compared to the ECH group (Figs. 3A and 3B). Additionally, except for the RANKL genes, whose expression was reduced and significantly different between the two groups, the messenger RNA (mRNA) expression levels of osteogenic-related genes (*RUNX*-2, *ALP*, *COL*, *OCN*, and *OPG*) manifested a slight increase but the difference was not statistically significant (Fig. 3C). In conclusion, CL-ECH and ECH shared a potential and similar ability to promote osteogenesis in BMSC cells.

## CL-ECH pretreatment inhibited RANKL-induced osteoclastogenesis in BMDMs

TRAP staining and ghost pen cyclic peptide staining were performed during RANKL-induced osteoclastogenesis, and the results demonstrated significantly inhibited osteoclastogenesis in both ECH and CL-ECH groups (Fig. 4A). Through the detection of osteoclast-related gene expression, we found that the mRNA expression levels of NF-$\kappa$B, C-FOS, and MMP-9 were further reduced in CL-ECH group but those of TRAP genes remained unchanged (Fig. 4B). This revealed that CL-ECH inhibited osteoclastogenesis from decreasing osteoclast inflammation levels.

## Evaluation of HE and MASSON tissue staining

Observation of the tissue sections under an inverted light microscope showed that compared to the control group (Fig. 5), the number of bone trabeculae was significantly reduced

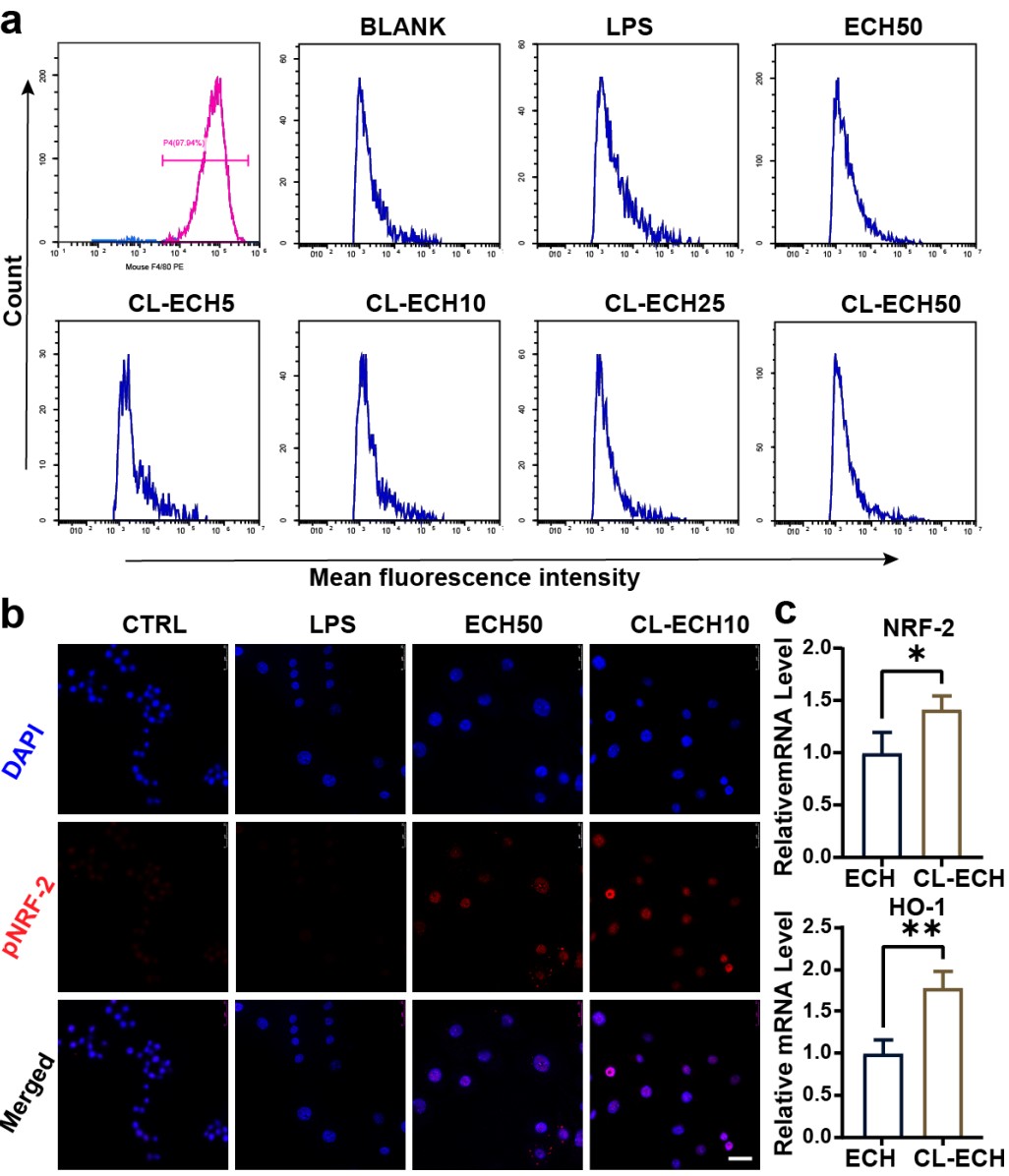

**Figure 2** **Antioxidant effect of the ECH or CL-ECH nanoparticles.** (A) Flow cytometry analysis showing identification of BMDMs and the effect of ECH (50 µM) or CL-ECH (5–50 µM) on ROS levels in BMDMs. (B) Immunofluorescence assay showing the expression of pNRF-2 in the BMDMs. (C) qRT-PCR showing the mRNA expression levels of NRF-2 and HO-1 in the BMDM. *$P < 0.05$, **$P < 0.01$. Scale bar: 25 µm. CL-ECH, cross-linker-echinacoside; BMDMs, bone marrow-derived macrophages; TEM, transmission electron microscopy; ROS, reactive oxygen species; qRT-PCR, quantitative reverse transcription PCR.

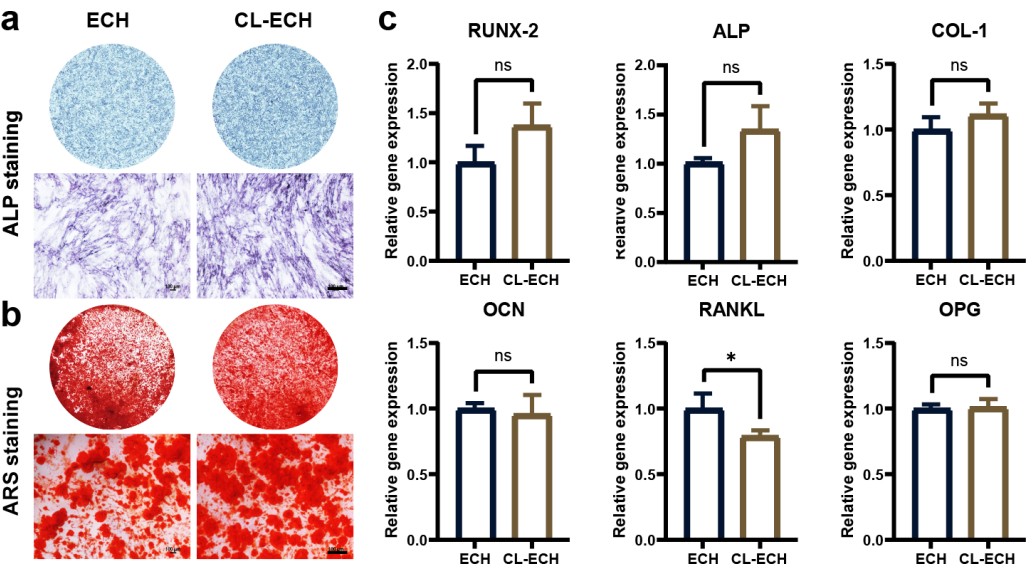

**Figure 3** **The effect of CL-ECH nanoparticles on the osteogenic ability of BMSCs was compared with of free ECH.** (A) ALP staining of the BMSCs showing ALP activity on day 7. (B) ARS staining of the BMSC showing mineralized nodules on day 21. (C) mRNA expression levels of genes (*RUNX-2*, *ALP*, *COL*-1, *OCN*, *RANKL*, and *OPG*) in the BMSCs, as determined by RT-PCR on day 5. *$P < 0.05$. Scale bar: 100 µm. CL-ECH, cross-linker-echinacoside; BMSCs, bone marrow stromal cells; ALP, Alkaline phosphatase; ARS, Alizarin Red S; RT-PCR, quantitative reverse transcription PCR.

and the distribution of bone trabeculae was sparser in the OVX group, indicating that the osteoporosis model was established successfully. After the ECH intervention, we also found that the number of trabeculae increased and the density increased and bone loss was inhibited in the ECH administration group. Interestingly, CL-ECH further reduced the loss of bone trabeculae.

Based on these findings, a molecular mechanism diagram was developed, as shown in Fig. S2. Cross-linked nanoparticles based on ECH had unique nano-biological effects by promoting the antioxidant effect and inhibiting osteoclast generation of macrophages regulated by delivered ECH.

## DISCUSSION

Antiresorptive drugs and anabolic agents are currently the two main categories of drugs used to treat osteoporosis (*Xu et al., 2022*), and all these drugs have different degrees of toxic side effects. Therefore, the development of natural drugs could focus on regulating bone metabolic state to restore the balance of bone metabolism.

In this study, we prepared CL-ECH nanoparticles, hoping to use this simple and effective drug carrier at low doses for the treatment of osteoporosis. Considering the molecular structure of ECH, the particle size of CL-ECH nanoparticles was optimized by adjusting the ratio of ECH and cross-linking agent, and finally, nanoparticles smaller than 100 nm in size were developed with a homogeneous spherical shape. This nano-releasing

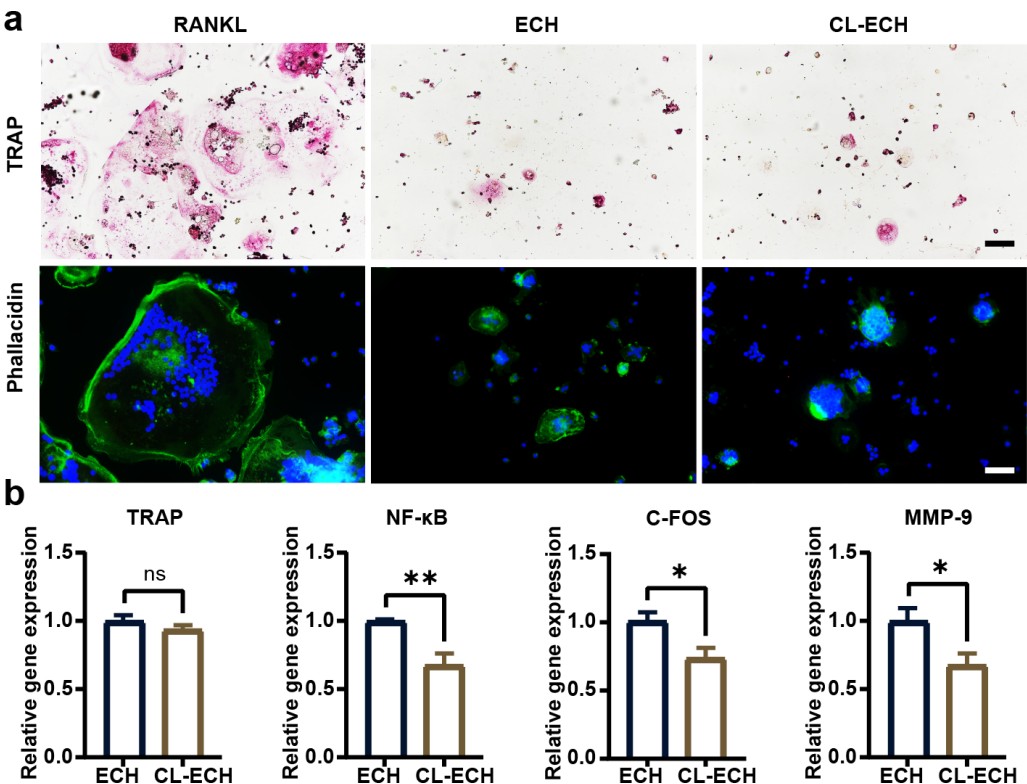

**Figure 4** **The effect of CL-ECH nanoparticles on osteoclast activities compared with that of free ECH.**
(A) TRAP and phalloidin staining showing an inhibitory effect of the ECH (50 µM) or CL-ECH (10 µM)
nanoparticles on the formation of RANKL-induced RAW264. 7 osteoclasts. (B) mRNA expression levels
of TRAP, NF-$\kappa$B, C-FOS, and MMP-9 in ECH or CL-ECH nanoparticle-treated osteoclasts on day 5 as
determined by qRT-PCR. *$P < 0.05$, **$P < 0.01$. Scale bar: 25 µm. CL-ECH, cross-linker-echinacoside;
TRAP, tartrate-resistant acid phosphatase; qRT-PCR, quantitative reverse transcription PCR.

body achieved a slow release in PBS solution in the presence of esterase and was effectively
taken up by cells.

The interaction between osteoblasts and immune cells in osteo-immunization has also
attracted much research attention (*Yao et al., 2021*; *Zhu et al., 2023*). In this study, three
types of cells (BMDM, BMSC, and RAW264.7) were selected to study the antioxidant
effect of CL-ECH nanoparticles on BMDM immune cells. We found that the antioxidant
effect of low-dose CL-ECH nanoparticles were significantly reduced on BMDM immune
cells, which was achieved by increasing the nuclear translocation of pNRF-2. This finding
was consistent with previous studies (*Kim et al., 2020*; *Xu et al., 2020*). Low-dose CL-ECH
nanoparticles had limited effect on the osteogenic ability of osteoblast-associated cells
(BMSCs) but notably reduced the expression of RANKL. This result was consistent with
the fact that RANKL-induced osteoclastogenesis in RAW264.7 cells was significantly
inhibited by CL-ECH nanoparticles.

NRF-2, an important member of the transcription factor family, plays an important
defense role in the complex regulatory network of immune inflammatory response and
is involved in the regulation of cancer development and progression (*Guo et al., 2015*).

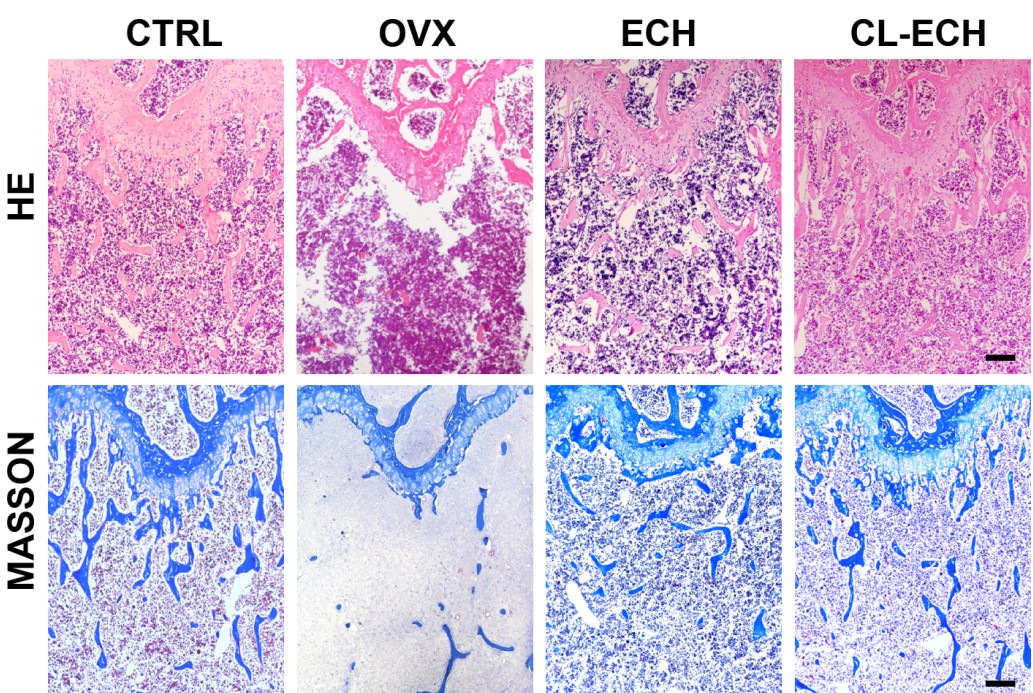

**Figure 5   HE and Masson staining of sectioned bone tissues of the mice in the groups. Scale bar: 100 μm.** CL-ECH, cross-linker-echinacoside.

Under normal physiological conditions, NRF-2 protein specifically binds to Keap-1 in the cytoplasm mainly through its N terminus. When NRF-2 is inactive in the cytoplasm, it is rapidly degraded through the ubiquitinated proteasome pathway. When cells are subjected to oxidative stress, NRF-2 uncouples from Keap-1, phosphorylates, and translocases into the nucleus, where it recognizes and binds to antioxidant response elements (AREs) to initiate transcription of the downstream target genes, such as antioxidant genes (*Ulasov et al., 2022*). The present study revealed that small cells can play an antioxidant and antitoxic role through molecular docking.

NRF-2 and P65 have been studied separately in previous study (*Abdelhamid et al., 2020*). In the current research, we offered a new perspective concerning the interaction of the two classical proteins, and discovered that the two proteins can dock together. NRF-2 may have a direct effect on P65, but whether there was a direct interaction between the two should be clarified by further research. Notably, it was found that the small molecule drug ECH could dock with NRF-2 by hydrogen bonding, and we hypothesized that ECH and Keap-1 competed for Keap-1, or NRF-2 uncoupled with Keap1, and then ECH bound to the NRF-2 protein, thereby increasing NRF-2 nuclear translocation and producing the antioxidant effect. Therefore, the loaded ECH nano-releasing body can significantly reduce the effective concentration of ECH and fulfill the desired antiosteoporosis effect.

In summary, our study preliminarily demonstrated that loaded ECH nanoreleasers were able to inhibit the oxidative stress of BMDMs by promoting NRF-2 nuclear translocation and effectively reduced the toxic effects resulted from high-dose ECH at the same time.

The current findings may provide a theoretical foundation for the development of novel anti-osteoporosis drugs. Nevertheless, this study still had some limitations remained to be further explored and improved. Firstly, the direct target of ECH was still not clear, which should be further confirmed by whole-genome sequencing. By doing so, it will also contribute to the development of new drugs and new targets in osteoclastic differentiation and functions.

## CONCLUSIONS

Compared to ECH, low-dose CL-ECH nanoparticles can be used to treat osteoporosis *in vivo*. *In vitro* experiments indicated that oxidative stress in BMDMs was inhibited through promoting NRF-2 nuclear translocation, which in turn suppressed the production of ROS and osteoclast production by downregulating NF-$\kappa$B expression, with limited effect on the osteogenesis of BMSCs. In conclusion, a low dose of natural drug-loaded ECH nanoparticles could serve as a promising drug for treating osteoporosis.

**Abbreviations**

| | |
|---|---|
| **ECH** | echinacoside |
| **CL** | cross-linker |
| **ROS** | reactive oxygen species |
| **BMDMs** | bone marrow-derived macrophages |
| **BMSCs** | bone marrow stromal cells |
| **qRT-PCR** | real-time quantitative polymerase chain reaction |
| **ALP** | Alkaline phosphatase |
| **AES** | Alizarin Red S |
| **OVX** | ovariectomized |
| **TNF$\alpha$** | tumor necrosis factor $\alpha$ |
| **TRAP** | Tartrate-resistant acid phosphatase |
| **DMEM** | Dulbecco's Modified Eagle Medium |
| **FBS** | fetal bovine serum |
| **TEM** | transmission electron microscopy |
| **UV** | ultraviolet |
| **cDNA** | complement DNA |
| **FITC** | fluorescein isothiocyanate |
| **HE** | hematoxylin and eosin |

### Funding

The study was funded by the National Natural Science Foundation of China (no. 81873715 and no. 82170913) and the Project of Shanghai Science and Technology Commission (no. 18441902100 and no. 201409006200). The funders had no role in study design, data collection and analysis, decision to publish, or preparation of the manuscript.

## Grant Disclosures

The following grant information was disclosed by the authors:
National Natural Science Foundation of China: 81873715, 82170913.
Project of Shanghai Science and Technology Commission: 18441902100, 201409006200.

## Competing Interests

The authors declare there are no competing interests.

## Author Contributions

- Dandan Hu conceived and designed the experiments, analyzed the data, authored or reviewed drafts of the article, and approved the final draft.
- Chunan Cheng conceived and designed the experiments, prepared figures and/or tables, and approved the final draft.
- Zhen Bian performed the experiments, authored or reviewed drafts of the article, and approved the final draft.
- Yubo Xu performed the experiments, analyzed the data, prepared figures and/or tables, and approved the final draft.

## Animal Ethics

The following information was supplied relating to ethical approvals (*i.e.*, approving body and any reference numbers):

The Ethics Committee of The Affiliated Stomatology Hospital of Tongji University provided full approval for this research (Approval number: [2022]- DW- 02).

## Data Availability

The raw data is available at GitHub and Zenodo:

- https://github.com/6xuyubo/My-Raw-Data.git

- 6xuyubo. (2023). 6xuyubo/My-Raw-Data: First release of my raw data (v.1.0.0). Zenodo. https://doi.org/10.5281/zenodo.10370202.

## Supplemental Information

Supplemental information for this article can be found online at http://dx.doi.org/10.7717/peerj.17229#supplemental-information.

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
