# Peer review of "The role of echinacoside-based cross-linker nanoparticles in the treatment of osteoporosis"

_PeerJ, doi:10.7717/peerj.17229_

## Round 0.1 · original submission · Major Revisions

After rigorous external review, three reviewers have provided their concerns and some suggestions for revision. Please respond to the reviewers' comments point by point as required.

Reviewer 1 ·

Basic reporting

no comment

Experimental design

no comment

Validity of the findings

no comment

Additional comments

The subject of this study is to reveal the therapeutic effects of echinocandin (ECH)-based cross-linker nanoparticles (CL-ECH) on osteoporosis, and it is a predominantly cell-based experiment. In this study, the antioxidant effect of CL-ECH was characterized by flow cytometry and molecular detection of the modulation of oxidative activity of CL-ECH particles on bone marrow-derived macrophages BMDMs. Next. The differences in the osteogenic effects of CL-ECH compared to ECH were compared. Finally, a mouse model was constructed to validate the therapeutic advantages of CL-ECH on osteoporosis, i.e., a lower dose of CL-ECH significantly improved the disease and bone trabecular defects in mice compared with ECH. In conclusion, the overall idea of this study is poorly organized and generally meets the requirements for publication, but the following issues still need to be addressed before publication:
1. When investigating the antioxidant effects of ECH or CL-ECH nanoparticles in this study, why were only two molecules, NRF2 and HO-1, detected, and can these two genes represent the whole antioxidant reaction process? Please explain.
2. It is suggested to improve the content of the Abstract, which does not clarify which cell types CL-ECH acts on, thus readers need to read the content of the latter part of the article in order to draw conclusions, which is not a good reading experience, and it is suggested to specify the cell types used in the experiments in order to enhance the readability.
3. Why did this study choose bone marrow-derived macrophages BMDMs as experimental subjects, there are other types of cells used to study osteoporosis, such as mesenchymal cells for osteogenic differentiation, so why focus on BMDMs, please give a rational explanation.
4. The Introduction section of this paper has clearly stated the therapeutic effect of ECH on osteoporosis and the inhibitory effect on reactive oxygen species, so why repeat the experiment with CL-ECH? Aren't these two compositionally identical? Please explain why and add a description in the original article.
5. There exist a number of studies utilizing cross-linking agents for the treatment of osteoporosis that may be relevant to the conduct of this study, so it is suggested that this be added to the introduction section.
6. The description of the results in Figure 3C states that there was no significant difference in the expression of RUNX-2, ALP, COL, OCN, and OPG genes between the CL-ECH and ECH treatment groups, does this mean that the results are not meaningful? The original conclusion is "CL-ECH and ECH produce a similar osteogenic capacity", but this does not mean that the results are meaningful, and it is suggested that the statement be revised.
7. What the elevated level of TRAP expression means for the treatment of osteoporosis is suggested to be described. Also, what conclusions are suggested by the fact that TRAP in osteoblasts is not affected by CL-ECH, please add in the description of the results.
8. Are these stains in Figure 4A shown at the same scale? The "RANKL" group is somewhat of an intracellular structure, but the original text does not give a description of it, so we suggest adding it.
9. It is suggested to add the advantages of CL-ECH over ECH in the Discussion section, rather than treating toxicities as a focused disadvantage of other drugs for osteoporosis.
10. This paper spends a lot of space focusing on the therapeutic mechanism and therapeutic effect of NRF-2, but there is another molecule HO-1 that is also tested in the results of this paper, so why is the role of this molecule not discussed? Please give a rational explanation and add a description if necessary.

Reviewer 2 ·

Basic reporting

In this study, Hu et al. prepared CL-ECH nanoparticles and found that it inhibited oxidative stress in BMDMs by promoting NRF-2 nuclear translocation, and then inhibited ROS production and osteoclast production by down-regulating NF-xB expression. It is also helpful in the treatment of osteoporosis in vivo at low doses.
1. In Line 67-79, the author describes the conception and experimental content of this paper. The author should summarize these contents instead of separating them into several paragraphs, which will better reflect the main idea of this study.
2. The full names of some abbreviations in the text are missing, such as ALP.
3. I suggest you have a colleague who is proficient in English and familiar with the subject matter review your manuscript, or contact a professional editing service.

Experimental design

4. The methods section lacks basic information about the materials or instruments used, such as the source and item number of LPS, iQ SYBR Green Supermix, ALP staining solution, the manufacturers of transmission electron microscopy, ultraviolet spectrophotometer.
5. In line 168, what is the calculation method of the drug loading rate, and what does the wt in 8.4 wt% refer to?
6. In line 172-173, “Flow cytometry analysis showed that Cy5-loaded CL-ECH nanoparticles were endocytosed into BMDMs, with around 15.06% of cells (and 9.02% of BMSC) being Cy5 positive.” The data described here do not correspond to those seen in Figure 1E.
7. The font in Figure 2A is too small, and we did not find "CL-ECH decreased ROS production in a dose-dependent manner" in this figure. Please supplement the statistical bar chart or statistical table of this result to show the credibility.
8. In line 191-192, the two groups are 50 uM ECH group and 10 uM CL-ECH, respectively, but the two groups compared in Figure 2C are ECH group and CL-ECH group.

Validity of the findings

no comment

Additional comments

9. The authors analyzed the effect of CL-ECH nanoparticles on osteoporosis in mice, and why the authors did not conduct Pharmacokinetic Study for CL-ECH nanoparticles in mice, which is also an important indicator to evaluate the safety and practicability of drugs.
10. In line 252, the authors mentioned that they discovered that the small molecule drug ECH could dock with NRF-2 by hydrogen bonding via molecular docking, However, the results section does not provide any pictures and descriptions of molecular docking.
11. Line 250-260, the authors collated all the results and extrapolated them in conjunction with other reports, but the organization was too fragmented and suggested that the authors make adjustments to increase readability.

·

Basic reporting

Basic Report:
The current study titled “The role of echinacoside-based cross-linker nanoparticles in the treatment of osteoporosis” has aimed to explore the the therapeutic effect of cross-linker (CL)-ECH 29 nanoparticles on osteoporosis compared to free echinacoside (ECH). The authors seem to have observed that in contrast to ECH, CL-ECH nanoparticles could inhibit oxidative stress in bone marrow derived macrophages (BMDMs) by promoting NRF-2 nuclear translocation. In the present scenario, the study is informative and might be interesting for the PeerJ readers. Also, the authors have used crisp and clear English to discuss their hypothsis and the figures are also upto the mark as per as publication standards are considered.

Experimental design

Experimental design:
The have done ample amount of experiments to support their hypothesis. However. have the authors explored the free radical scavenging capacity of Echinocandin-based CL-ECH nanoparticles. If not, the authors can consider conducting this experiment. Also, the extent of their interaction with individual free radicals can be studied in details. This would greatly contribute to the overall quality of the manuscript.

Validity of the findings

Validity of the Findings:
In this context the authors are advised to highlight the novelty of their study, in details, in the Introduction section. The authors need to elaborate on the future perspectives of their findings, in details, at the end of their Discussion.

Additional comments

Minor Suggestions:
1. The authors can consider adding a list of abbreviations, for better readability of the readers.
2. The authors have not prepared a well designed graphical abstract and will help the readers understand their findings.
3. The authors should consider adding the DOI numbers of the references cited, wherever available.

---

## Round 0.2 · accepted · Accept

Based on the feedback from both reviewers, it is clear that the authors have appropriately addressed all of the comments and suggestions provided. The manuscript has been revised and improved in response to the reviewers' feedback, strengthening the overall quality and clarity of the research presented. Therefore, I am pleased to confirm that this manuscript is now ready for publication.

Reviewer 1 ·

Basic reporting

no comment

Experimental design

no comment

Validity of the findings

no comment

Additional comments

The manuscript titled "The role of echinacoside-based cross-linker nanoparticles in the treatment of osteoporosis" presents a study exploring the therapeutic effects of echinacoside (ECH)-infused cross-linker (CL) nanoparticles compared to ECH alone in the context of osteoporosis treatment. Echinacoside is highlighted as a natural, small molecule drug with potential benefits for treating osteoporosis due to its antioxidant and anti-inflammatory properties. The research investigates the efficacy of these nanoparticles in both in vitro and in vivo settings, focusing on their ability to suppress oxidative stress in bone marrow-derived macrophages (BMDMs), enhance the osteogenic capacity of bone marrow stromal cells (BMSCs), and inhibit osteoclast production.
This manuscript presents a comprehensive and well-structured study that addresses a significant gap in osteoporosis treatment. The innovative approach of using echinacoside-based cross-linker nanoparticles showcases the potential for developing more effective and less toxic treatments. The detailed methodology and clear presentation of results contribute to the manuscript's strengths. However, further research might be necessary to fully understand the long-term effects and safety profile of these nanoparticles in humans. The study is a valuable addition to the field of osteoporosis research, with its findings laying the groundwork for future investigations into novel treatments.

Reviewer 2 ·

Basic reporting

no comment

Experimental design

The study proposes an innovative approach to osteoporosis treatment using echinacoside (ECH)-based cross-linker (CL) nanoparticles. The research question is well-defined, relevant, and addresses a significant gap in the current treatment modalities for osteoporosis, specifically the adverse side effects associated with current drugs.

Validity of the findings

The results section clearly demonstrates the beneficial effects of CL-ECH nanoparticles on suppressing oxidative stress in BMDMs and promoting NRF-2 nuclear translocation without significantly affecting the osteogenesis of BMSCs. In vivo studies further validate the potential of low-dose CL-ECH nanoparticles in improving bone trabecular loss compared to ECH administration alone. The conclusions drawn are well-supported by the data presented.

Additional comments

The manuscript presents a well-designed and executed study on the development and assessment of ECH-based cross-linker nanoparticles for osteoporosis treatment. The experimental design is rigorous, and the methods are robust, ensuring reproducibility. Results are analyzed appropriately, and conclusions are supported by the data. The authors also appropriately address limitations and future research directions, contributing valuable insights into potential new treatments for osteoporosis.